



# Parametric resonance in the dynamics of an elliptic vortex in a periodically strained environment

Konstantin V. Koshel[1] and Eugene A. Ryzhov[1]

[1]V.I.Il'ichev Pacific Oceanological Institute, 43, Baltiyskaya Street, Vladivostok, 690041, Russia

*Correspondence to:* K.V. Koshel
(kvkoshel@poi.dvo.ru)

**Abstract.** The model of an elliptic vortex evolving in a periodically strained background flow is studied in order to establish the possible unbounded regimes Depending on the parameters of the exterior flow, there are three classical regimes of the elliptic vortex motion under constant linear deformation: (i) rotation, (ii) nutation, and (iii) infinite elongation. The phase portrait for the vortex dynamics features critical points, which correspond to the stationary vortex not changing its form and orientation. We demonstrate that, given superimposed periodic oscillations to the exterior deformation, the phase space region corresponding to the elliptic critical point experiences parametric instability leading to locally unbounded dynamics of the vortex. This dynamics manifests itself as the vortex nutates along the strain axis while continuously elongating. This motion continues until nonlinear effects intervene near the region associated with the steady-state separatrix. Next, we show that for specific values of the perturbation parameters, the parametric instability is effectively suppressed by nonlinearity in the primal parametric instability zone. The secondary zone of the parametric instability, on the contrary, produces an effective growth of the vortex's aspect ratio.

## 1 Introduction

Simplified vortex models have been extremely useful in helping us understand the intricate behavior of real coherent structures in the ocean and atmosphere. Such models are usually highly nonlinear making it possible to get insight into many hardly predictable phenomena taking place in the geophysical media (Provenzale, 1999; Balasuriya and Jones, 2001; Kostrykin et al., 2006; Koshel and Prants, 2006; Koshel et al., 2008; Samelson, 2013; Koshel et al., 2013; Ryzhov and Koshel, 2013; Koshel et al., 2014; Haller, 2015). For instance, such vortex models can shed some light into the dynamics of interacting coherent mesoscale vortices (Reznik and Dewar, 1994; Gryanik et al., 2000; Reznik and Kizner, 2010; Carton et al., 2010, 2013; Reinaud and Carton, 2015), sustainability of such vortices against external flows McKiver and Dritschel (2003); Liu and Roebber (2008); Perrot and Carton (2010) or topographic influence (Kozlov et al., 2005; Johnson and McDonald, 2005; Ryzhov et al., 2010; Sutyrin et al., 2011; Nilawar et al., 2012) and so on.

One of the most renowned vortex models is the model of an elliptic vortex subjected to linear deformation consisting of shear and rotational components (Kida, 1981). In the case of stationary deformation, the model permits an elliptic vortex to perform three types of motion (Kida, 1981) depending on the parameters of the deformation flow and the initial alignment of the ellipse

against the exterior strain. There are two periodic states involving the vortex changing its eccentricity, these are (i) rotation, and (ii) nutation. Moreover, there is one aperiodic state – infinite elongation. In this case, the vortex elongates continuously tending to be collinear with the strain axis. Moreover, for a specific initial alignment, the vortex can be stationary not performing any

motion.

The model of an elliptic vortex embedded in a linear deformation field is the base model to assess the stability of elliptic vortex shapes occurring in nature. A large body of literature is devoted to the problem. Most of the papers consider spatial perturbations to the elliptic form in the case of constant linear deformation (Neu, 1984; Melander et al., 1986; Neu, 1990; Dritschel, 1990; Meacham et al., 1990; Legras and Dritschel, 1991; Kida and Takaoka, 1994; Miyazaki and Hanazaki, 1994;

Bayly et al., 1996; Meacham et al., 1997; Mitchell and Rossi, 2008). A prominent result of this stability analysis is that an elliptic vortex is stable to linear perturbations of its form until its geometrical shape complies with the relation $a/b \leq 3$, where $a$ and $b$ are the major and minor semi-axes of the ellipse.

Another interesting aspect of the dynamics of an elliptic vortex is its response to time-dependent external deformation. In this case, the form of the vortex endures no changes but the vortex's aspect ratio and orientation can alter unpredictably

demonstrating chaotic behavior. This problem has been addressed for small-amplitude strain oscillations by means of the Melnikov's integral technique in the works (Bertozzi, 1988; Dhanak and Marshall, 1993; Ide and Wiggins, 1995; Goldman and McCann, 2008). The evolution of the vortex embedded in a time-dependent strain with a slowly varying frequency was addressed in (Friedland, 1999). The authors of the paper (Dhanak and Marshall, 1993) also assess the stability of the stationary configuration of the elliptic vortex, i.e. the vortex's aspect ratio and orientation do not change in time under a constant exterior

deformation, given a time-dependent external perturbation. They show that such a stationary configuration of the vortex can be easily destabilized into the nutation or even rotation regimes because of a linear resonance effect. In this paper, we demonstrate that, in the case of relatively small amplitudes of the perturbation, the vortex loses its stationarity by means of parametric instability. In the case of finite amplitudes and optimal frequencies of the perturbation, the dynamics of the vortex is governed strictly by nonlinear effects.

It is also worth noting that fluid particle advection near an oscillating elliptic vortex embedded in a constant deformation flow manifests chaotic dynamics (Polvani and Wisdom, 1990; Polvani et al., 1990; Dahleh, 1992). This is because the oscillating elliptic vortex generates a time-periodic perturbation to the fluid particle motion. This results in the appearance of exponentially diverging trajectories in the unsteady velocity field governing the fluid particle advection. Similar chaotic dynamics is present in the case of the ellipsoid vortex model (Zhmur et al., 2011; Koshel et al., 2013, 2015), which is a generalization of the elliptic

vortex model taking into account a linear vertical stratification of the baroclinic external flow (Zhmur and Pankratov, 1989; Meacham et al., 1994; Dritschel, 2011; McKiver, 2015; McKiver and Dritschel, 2016).

Let us consider an inviscid, incompressible, two-dimensional flow. In this flow, an elliptic patch of constant vorticity $g$, experiencing deformation from time-dependent strain $e(t)$ and background rotation $\gamma(t)$, is embedded. The patch conserves its elliptic form with $a$, and $b$ being the ellipse's semi-axes, $\varepsilon = a/b$ being the aspect ratio and $\varphi$ being the angle between the





ellipse's major semi-axis and the x-axis of the Cartesian coordinate frame. The governing equations are (Kida, 1981)

$$\dot{\varepsilon} = 2e\varepsilon\cos 2\varphi, \quad \dot{\varphi} = \gamma + \frac{g\varepsilon}{(\varepsilon+1)^2} - e\frac{\varepsilon^2+1}{\varepsilon^2-1}\sin 2\varphi. \tag{1}$$

It is worth noticing that the motion of the ellipse's center $(x_0, y_0)$ is governed by advection equations in the form

$$\frac{dx_0}{dt} = u_0 + e(x_0 - x_d) - \gamma(y_0 - y_d),$$

$$\frac{dy_0}{dt} = v_0 - e(y_0 - y_d) + \gamma(x_0 - x_d), \tag{2}$$

where $u_0, v_0$ are the velocity's components of an arbitrary uniform flow, $x_d, y_d$ are the coordinates of the constant deformation center. The equations (2) exactly coincide with the equations governing the motion of the vorticity center of an arbitrary number of point vortices embedded in such a deformation flow (Koshel and Ryzhov, 2012; Ryzhov and Koshel, 2016). Given periodic unequal dependencies of the strain and rotation components, these equations are allowed for parametric instability. This, in turn, may result in unbounded motion of the vorticity center depending on the strain and rotation oscillation parameters. However, the relative vortex motion is independent of the type of the vortex's center motion. Further, we consider the evolution of the elliptic vortex when the vortex's center and the exterior deformation's center coincide at the coordinate's origin.

Without loss of generality, the ellipse's vorticity in eq. 1 is further set to be $g = 1$. The equations (1) represent a dynamical system with 'one and a half' degrees of freedom (Lichtenberg and Lieberman, 1983; Zaslavsky, 1998) provident time-dependent strain and rotation rates $e(t)$ and $\gamma(t)$. This suggests that the dynamics of the phase variables may be chaotic for certain initial conditions. To start, it is informative to look into the stationary system given constant values of $e(t) \equiv e_0 = const$, $\gamma(t) \equiv \gamma_0 = const$. The critical points corresponding to the stationary elliptic vortex ensue from the relations (Kida, 1981; Bayly et al., 1996)

$$\varphi_0 = \pm\frac{\pi}{4},$$

$$\varepsilon_0{}^3 + \varepsilon_0{}^2 \frac{(\gamma - e\sin 2\varphi_0 + 1)}{(\gamma - e\sin 2\varphi_0)} - \varepsilon_0 \frac{(\gamma + e\sin 2\varphi_0 + 1)}{(\gamma - e\sin 2\varphi_0)} - \frac{(\Omega + e\sin 2\varphi_0)}{(\Omega - e\sin 2\varphi_0)} = 0. \tag{3}$$

There are four qualitatively different phase portraits depending on the number and type of the critical points (Bayly et al., 1996). Each phase portrait has only one elliptic critical point. The corresponding phase portrait for the values $e_0 = 0.15$, $\gamma_0 = 0.02$ is shown in fig.1. The homoclinic separatrix delineates the regions of the initial conditions corresponding to the ellipse nutation (near the elliptic critical point) and infinite elongation (all the rest). The homoclinic separatrix delineates the regions of the initial conditions corresponding to the ellipse nutation (near the elliptic critical point) and infinite elongation (all the rest).

The elliptical critical point illustrated in fig. 1 corresponds to the elliptic vortex not performing any motion. Once slightly shifted from this stance, the vortex starts nutating slightly changing the lengths of its semi-axes and the orientation angle. If an initial condition is taken outside of the nutation region near the elliptic critical point, the vortex starts elongating infinitely.





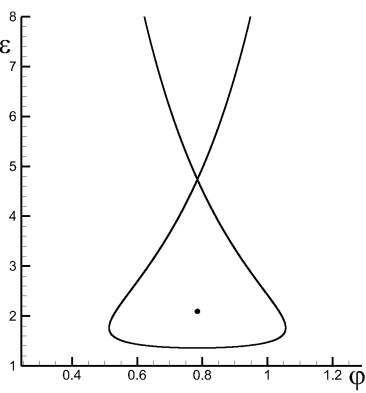

**Figure 1.** The phase portrait of the stationary system (1) in the case of one elliptic and one hyperbolic critical points for $e = 0.15$, $\gamma = 0.02$.

## 2    The dynamics of the perturbed system near the stationary position

Now, let us examine the dynamics of the system near the elliptic critical point given small time-dependent perturbations of the deformation flow. One can expand eqs. (1) into a Taylor series up to the first term over a small deviation from the elliptic point

$$\varepsilon\left(t\right) \approx \varepsilon_0 + \varepsilon'\left(t\right), \ \ \varphi\left(t\right) \approx \varphi_0 + \varphi'\left(t\right) = \pm\frac{\pi}{4} + \varphi'\left(t\right), \tag{4}$$

where $\varepsilon_0, \varphi_0$ follow from (3), $\varepsilon'\left(t\right), \varphi'\left(t\right)$ are small time-dependent deviations. One then obtains for the deviations

$$\frac{d\varepsilon'}{d\tau} = -4e\left(t\right)\varepsilon_0\varphi'\sin 2\varphi_0,$$

$$\frac{d\varphi'}{d\tau} = \left[\gamma\left(t\right) - e\left(t\right)\frac{\varepsilon_0{}^2 + 1}{\varepsilon_0{}^2 - 1}\sin 2\varphi_0\right] - \varepsilon'\left[\frac{\left(\varepsilon_0 - 1\right)}{\left(\varepsilon_0 + 1\right)^3} - 4e\left(t\right)\frac{\varepsilon_0}{\left(\varepsilon_0{}^2 - 1\right)^2}\sin 2\varphi_0\right]. \tag{5}$$

The linear system (5) is an inhomogeneous one. However, in terms of determining whether the system has unbounded solutions, one can take advantage of the associated homogeneous system

$$\frac{d\varepsilon'}{d\tau} = -4e\left(t\right)\varepsilon_0\varphi'\sin 2\varphi_0,$$

$$\frac{d\varphi'}{d\tau} = -\varepsilon'\left[\frac{\left(\varepsilon_0 - 1\right)}{\left(\varepsilon_0 + 1\right)^3} - 4e\left(t\right)\frac{\varepsilon_0}{\left(\varepsilon_0{}^2 - 1\right)^2}\sin 2\varphi_0\right]. \tag{6}$$

It is worth noting that the term with the background rotation $\gamma$ has vanished from the homogeneous system. This readily means that the existence of unbounded solutions about the stationary position is independent on the exterior rotation.

Now, let $e(t) = \delta\cos\nu t$ be the time-dependent strain. One can see that if reduced to a second order equation, the system (6) represents a Hill equation (Magnus and Winkler, 1966). This, in turn, signifies that there is possible the manifestation of parametric instability near the vortex stationary position. In other words, given specific values of the perturbation's parameters, the phase trajectories become unbounded near the steady-state elliptic critical point. The values of the perturbation parameters



resulting in parametric instability can be readily figured out using the Floquet analysis. An analytical estimate can be derived by means of averaging techniques (Klyatskin and Koshel, 1983; Koshel and Ryzhov, 2012, 2016; Ryzhov and Koshel, 2016). To do this, it is convenient to rewrite the system 6 in the form

$$
\frac{d\varepsilon'}{dt} = -4e\left(t\right)\varepsilon_0 \varphi' \sin 2\varphi_0,
$$

$$
\frac{d\varphi'}{dt} = \left(\frac{k^2}{4e_0\varepsilon_0 \sin 2\varphi_0} + 4\frac{\varepsilon_0 \sin 2\varphi_0}{\left(\varepsilon_0^2 - 1\right)^2}\delta \cos \nu t\right)\varepsilon',
\tag{7}
$$

where $k^2 = -\frac{4e_0\varepsilon_0 \sin 2\varphi_0}{(\varepsilon_0+1)^2}\left[\frac{(\varepsilon_0-1)}{(\varepsilon_0+1)} - 4\frac{e_0\varepsilon_0}{(\varepsilon_0-1)^2}\sin 2\varphi_0\right].$

Let us introduce a new variable, $\rho = -\left(1 + i\frac{4e_0\varepsilon_0 \sin 2\varphi_0}{k}\frac{\varphi'}{\varepsilon'}\right)e^{-i\nu\tau}$. Then, instead of (7), one obtains

$$
\frac{d\rho}{dt} = i\left(2k\left[1 + \frac{\delta\left(e^{i\nu t} + e^{-i\nu t}\right)}{2e_0}\right] - \nu\right)\rho
$$

$$
-ik\left[\frac{16}{k^2}\frac{e_0{}^2\varepsilon_0{}^2}{\left(\varepsilon_0^2-1\right)^2} - 1\right]\frac{\delta\left(1 + e^{-2i\nu t}\right)}{2e_0} + ik\left[e^{i\nu t} + \frac{\delta\left(e^{2i\nu t}+1\right)}{2e_0}\right]\rho^2.
\tag{8}
$$

Then, by omitting the fast-oscillating terms, one gets for the averaged value $\bar{\rho}$

$$
\frac{d\bar{\rho}}{dt} = ik\frac{\delta}{2e_0}\bar{\rho}^2 + i\left(2k - \nu\right)\bar{\rho} - ik\left[\frac{16}{k^2}\frac{e_0{}^2\varepsilon_0{}^2}{\left(\varepsilon_0^2-1\right)^2} - 1\right]\frac{\delta}{2e_0},
$$

$$
\bar{\rho}\left(0\right) = 0.
\tag{9}
$$

The solution of (9) is

$$
\frac{\left(\bar{\rho} + \frac{e_0}{\delta k}\left(2k - \nu\right) - D\right)}{\left(\bar{\rho} + \frac{e_0}{\delta k}\left(2k - \nu\right) + D\right)} = \exp\left\{ik\frac{D\delta}{e_0}\right\},
\tag{10}
$$

where $D^2 = \left(\frac{e_0}{\delta k}\left(2k - \nu\right)\right)^2 - \left[\frac{16}{k^2}\frac{e_0{}^2\varepsilon_0{}^2}{(\varepsilon_0+1)^2(\varepsilon_0-1)^2} - 1\right].$

The solution (10) grows unboundedly if the exponent in the right-hand term exponent function is real. Therefore, a rough analytical estimate for the location of the primary parametric instability zone in the parametrci space ensues

$$
\left(2k - \nu\right) = \pm 2\delta\sqrt{\frac{\sin 2\varphi_0}{e_0}\frac{\varepsilon_0\left(\varepsilon_0 - 1\right)}{\left(\varepsilon_0 + 1\right)^3}}.
\tag{11}
$$

Figure 2 depicts the precise parametric instability zones obtained by the Floquet analysis (the dark regions) and the analytically estimated values (the dashed line) in the $(\delta, \nu)$ parametric space. Every parameter set from the dark regions leads to spiral-like unbounded trajectories of the linear system (6). While the parameters taken from the light areas result in periodic trajectories in bounded regions.



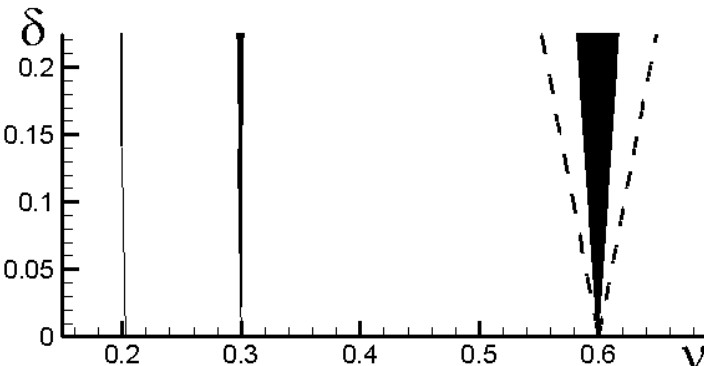

**Figure 2.** The regions of parametric instability in the parametric plane $\nu, \delta$. The dark regions correspond to locally unbounded dynamics of the ellipse obtained with the use of the linear approximation 6. The dashed lines mark the region incurred from the analytical estimate 11.

## 3   Nonlinear suppression of the solution growth attributed to the linear parametric instability

Now, let us discuss possible implications of the parametric instability reported for the linear system 6 that governs the dynamics of the elliptic vortex in the immediate vicinity of the elliptical critical point shown in fig. 1. Since the original system 1 is nonlinear, it is clear that the applicability range of the linear system 6 is relatively limited.

5      First of all, in the case of parametric instability, the trajectories originated near the steady-state elliptic critical point move unboundedly only up to the steady-state separatrix region, where nonlinear effects prevail the linear unbounded motion. This scenario realizes only if the nonlinear effects do not already drastically influence the dynamics in the very vicinity of the steady-state critical elliptic point. This is the case, for instance, for the primary parametric instability zone shown in fig. 2. Given the corresponding perturbation values, the phase space near the steady-state critical elliptical point differs crucially from the unperturbed phase space. The Poincaré section shown in fig.3a confirms this effect. The dynamics near the steady-state elliptic critical point is dominated by a nonlinear resonance with winding number $1 : 2$. Because of this effect, the phase trajectories originated near the steady-state elliptic critical point do not demonstrate parametric instability. The trajectory shown in fig. 3b is clearly bounded to the region of the nonlinear resonance influence. Therefore, the linear system 6 cannot account for the dynamics in this case.

15     Nevertheless, the linear system is a good approximation for certain values of the perturbation's parameters. For instance, when one considers the second parametric instability zone, the perturbed phase space near the steady-state elliptic critical point features no nonlinear resonances (see the Poincaré section shown in fig. 4a). Therefore, the dynamics near the steady-state critical elliptic point can be derived from the linear system 6. When this happens, the phase trajectory that starts near the steady-state critical elliptic point moves in a spiral-like divergent trajectory (see fig. 4b).





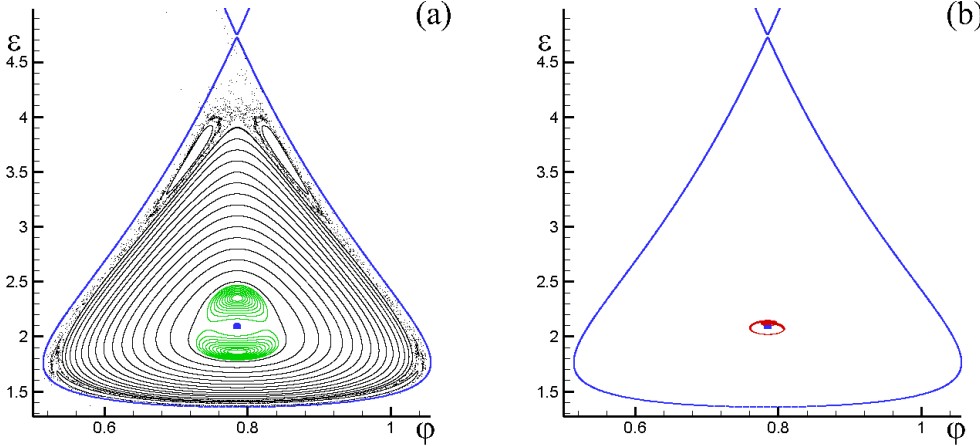

**Figure 3.** The dynamics of the perturbed system in the case of the primary parametric instability zone of the corresponding linearized system (6) for $\delta = 0.01$, $\nu = 0.6$. The solid blue lines show the steady-state separatrix. The blue dot shows the steady-state elliptic critical point. (a) a Poincaré section illustrating the appearance of a highly nonlinear zone (the green orbits) near the steady-state elliptic critical point that prohibits the divergent motion due to the linear parametric resonance; (b) a phase trajectory bounded to the region of the nonlinear resonance illustrating a vanishing effect of the linear parametric resonance.

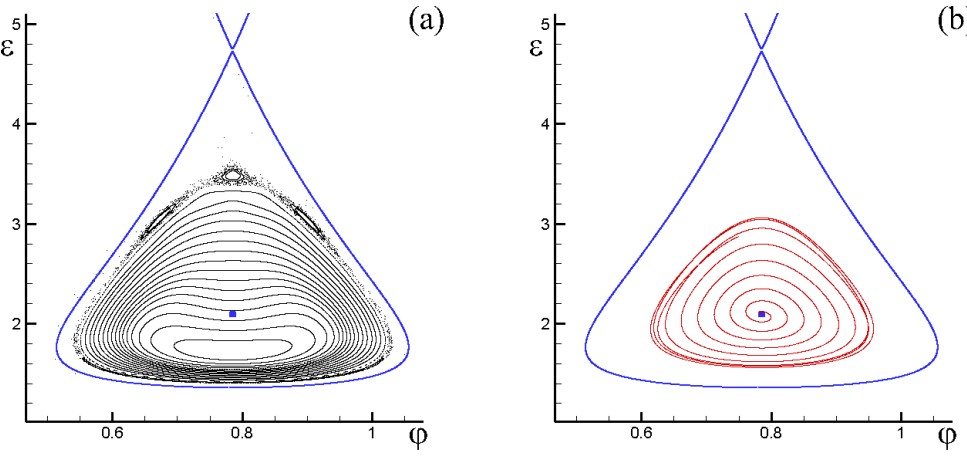

**Figure 4.** The same as in fig. 4 except $\nu = 0.3$. (a) a Poincaré section illustrating that the perturbed system remains largely linear near the steady-state elliptic critical point; (b) a phase trajectory experiencing linear parametric instability that results in a divergent spiral-like trajectory.





## 4  Conclusions

In this work, we have considered a model of an isolated elliptic vortex embedded in a time-dependent deformation flow, consisting of strain and rotational components. The main focus is on the dynamics of the vortex near its steady-state stable position. In the linear approximation, the vortex dynamics is established to experience parametric instability. This instability

entails the following motion of the vortex. The vortex starts nutating about the strain axis continuously increasing the aspect ratio of its semi-axes. This evolution continues until the phase trajectory of the vortex dynamics reaches the region of high nonlinearity near the steady-state separatrix. This regime proceeds provided the nonlinear system governing the dynamics of the vortex remains relatively linear near the stead-state elliptic point being driven. We have shown that this, in fact, is possible for certain values of the perturbation parameters. Thus, our results suggest that the parametric instability, which is intrinsic to

linear systems, can also be an important factor in the nonlnear system in question.

However, the influence of linear effects on the nonlinear systems should be generally interpreted with great care. Indeed, the primary parametric resonance zone produces the most effective unbounded motion of the phase trajectories in the linearized system, but the same driving parameters applied to the nonlinear system lead only to a bounded motion in a very small vicinity of the steady-state elliptic critical point. This is because a region of high nonlinearity appears near the steady-state elliptic

critical point that completely suppresses the linear unbounded motion in the nonlinear system.

Considering the evolution of the elliptic vortex, it means that the elliptic vortex nutates along the shear axis becoming more oblong in time. This dynamics is regular meaning that the vortex form returns to its initial shape after the unbounded motion is suppressed by the nonlinear effects near the steady-state separatrix. However, during this temporal elongation, the aspect ratio of the vortex may easily exceed the critical value of the vortex stability. Thus, in terms of the linear stability of the vortex

to the perturbations of its elliptic form, the vortex turns unstable. Therefore, our results may suggest the following scenario of the evolution of an elliptic vortex in a time-dependent natural environment. First, it starts becoming more and more oblate due to the linear parametric instability. After attaining the critical aspect ratio, small-scale disturbances start transpiring on the boundary of the vortex. Finally, it may break into a number of smaller elliptic vortices Carton et al. (1989); Polvani and Carton (1990); Carton and Legras (1994).

*Acknowledgements.*  The publication of this paper is made possible thanks to the Office of Naval Research Grant No. $N00014-16-1-2492$ The work of EAR in calculating the parameters leading to instability was partially supported by the Ministry of Education and Science of Russian Federation, project no. $MK-3084.2015.1$ and by the Russian Foundation for Basic Research, project no. $15-05-00103$. The work of KVK in obtaining the analytical estimate was supported by the Russian Scientific Foundation, project no. $16-17-10025$.

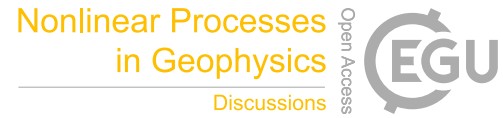

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
