# Peer review of "Parametric resonance in the dynamics of an elliptic vortex in a periodically strained environment"

_Nonlinear Processes in Geophysics, 2016_

## Referee Comment (RC1) · Anonymous Referee #1 · 4 Oct 2016

this paper is interesting, well designed and well written ; it should be published

I have only technical remarks which should be taken into account before the paper is printed

page 3 lines 24-26 repeat exactly the previous sentence at lines 23-24 - they should be suppressed

page 4 line 15 there is A POSSIBLE MANIFESTATION of parametric instability

page 5 line 17 in the parameTRIC space

page 6 lines 2, 4, 13 : linear system (6) – between parentheses

page 6 line 6 : prevail AGAINST the linear unbounded motion.

page 8 line 8 STEADY (a Y is missing) STATE ELLIPTIC POINT UNDER PERTURBATION

page 8 line 10 in the NONLINEAR system in question.

page 8 lines 23-24 please put the references between parentheses

---

## Referee Comment (RC2) · Anonymous Referee #2 · 12 Oct 2016

The short paper proposes an exploration of the effect of an oscillating external flow on a two-dimensional elliptical vortex vortex patch. In particular the effects of nonlinear suppression of the parametric instability growth on a couple of examples.

I believe the paper is interesting and overall well-written. I believe also that most of the results are original and can be accepted for publication with minor corrections (mostly typographical errors).

Minor points:

1) P5l l.10. 'omitting the fast-oscillating term ...' Why can the author do this? Does this term average to 0? Are the authors making a fast-time/slow-time separation?

[Figure]

2) Fig 3 and 4 should be more explained in the text, and caption should provide more information:

Questions which come to mind immediately:

i) In fig 3 and 4: are e=0.15 and \gamma = 0.02 from fig 1 still used? The same question goes for fig 2 in fact.

ii) Fig 3,4 a) Can the authors add a short sentence provides the details on how are in practice they obtained their Poincaré sections?

iii) Fig 3,4 b) What is the exact starting point of the trajectories used to illustrate the generic behaviour?

iv) It is unclear visually whether the trajectory in Fig 4b keeps spiralling outward for long times.

Minor points, typographical errors:

o) Abstract: add a full stop at the end of the first sentence.

a) p3, l1 "x-axis" -> "$x$-axis'

b) p3, eqn (3) \Omega seems undefined in the present paper. The author should not expect the reader to read Bayly et al (1996) to understand symbols.

c) p5, l7 What is $\tau$. It is a rescaled time $t$ or just $t$?

d) p5, 16. The sentence unclear. Maybe rephrase as "...if the argument in the right-handed exponential function.." Then, on the next line, typo : "parametrci" -> "parametric"

e) p6. l9 & l16. Be more specific when refefring to "primary" and "secondary" zones. What is meant? I guess primary is the zone around \nu = 0.6 and the secondary the one around \nu = 0.3 but it is unclear.

f). p7 caption of figure 4: "The same as in fig. 3" (not 4).

g) p8. l10 typo " nonlnear" -> "nonlinear".

---

## Referee Comment (RC3) · Anonymous Referee #3 · 17 Oct 2016

**Referee report for "Parametric resonance in the dynamics of an elliptic vortex in a periodically strained environment"**

This work considers the stability for an elliptic vortex subjected to an external straining flow, examing different dynamic regimes depending on the external flow parameters. This is a novel work, very much of interest to and worthy of publication in this journal. I find no problem technically with the paper, only minor typos, as well as some sentences which should be changed to provide more clarity.

**Minor Comments**

[Figure]

(1) page 1, Abstract, line 2: missing full stop after the word "regimes".

(2) page 1, line 15: Suggest changing wording to "Such models are usually highly nonlinear making it possible to gain insight into many phenomena that are difficult to predict within a geophysical setting"

(3) page 1, line 24: Suggest changing wording to "In the case of a stationary deformation, the elliptical vortex is able to perform..."

(4) page 2, line 3: Suggest to remove redundant phrase "not performing any motion", or else add comma ',' before the word not.

(5) page 3, line 9: Change to "these equations allow for parametric instability."

(6) page 3, line 11: Remove words "type of the"

(7) page 3, line 14: Unclear what is the intended meaning of the line "provident time-dependent strain and rotation rates...", please clarify

(8) page 3, line 25: Typo, remove repeated line "The homoclinic separatrix...."

(9) page 4, line 13: change word "on" to "of"

(10) page 5, line 17: Typo, change "parametrci" to "parametric"

(11) page 6, line 6: add word "over" after prevail

(12) page 6, line 7: change "realizes" to "is realized"

(13) page 6, line 7: change "very" to "immediate"

(14) page 8, line 8: Typo, change "stead" to "steady"

---

## Referee Comment (RC4) · Anonymous Referee #2 · 31 Oct 2016

The authors have taken all my comments into consideration and have amended the paper accordingly. As I mentioned in my first comments, I believe that this work should be published.

---

## Referee Comment (RC5) · Anonymous Referee #3 · 31 Oct 2016

The authors have made all the suggested corrections and now I accept the paper.

---

## Referee Comment (RC6) · Anonymous Referee #1 · 31 Oct 2016

the authors have taken my remarks into account and the paper should now be published

---

## Author Comment (AC1) · 31 Oct 2016

Response to Reviewer 1's comments on the paper "Parametric resonance in the dynamics of an elliptic vortex in a periodically strained environment" by K.V. Koshel and E.A. Ryzhov.

We thank the reviewer for commenting on the manuscript. We have addressed the issues raised by the reviewer. The following is the point-by-point responses to the reviewer's list of questions:

Reviewer's comments:

this paper is interesting, well designed and well written ; it should be published

I have only technical remarks which should be taken into account before the paper is printed

page 3 lines 24-26 repeat exactly the previous sentence at lines 23-24 - they should be suppressed

page 4 line 15 there is A POSSIBLE MANIFESTATION of parametric instability

page 5 line 17 in the parameTRIC space

page 6 lines 2, 4, 13 : linear system (6) – between parentheses

page 6 line 6 : prevail AGAINST the linear unbounded motion

page 8 line 8 STEADY (a Y is missing) STATE ELLIPTIC POINT UNDER PERTURBATION

page 8 line 10 in the NONLINEAR system in question

page 8 lines 23-24 please put the references between parentheses

Author's Response: All is corrected. Thank you very much for pointing these shortcomings out.

We again thank the reviewer for the recommendations.

Best wishes, Eugene A. Ryzhov, Konstantin V. Koshel

---

## Author Comment (AC2) · 31 Oct 2016

Response to Reviewer 2's comments on the paper "Parametric resonance in the dynamics of an elliptic vortex in a periodically strained environment" by K.V. Koshel and E.A. Ryzhov.

We thank the reviewer for commenting on the manuscript. We have addressed the issues raised by the reviewer. The following is the point-by-point responses to the reviewer's list of questions:

*Reviewer's comments:*
*The short paper proposes an exploration of the effect of an oscillating external flow on a two-dimensional elliptical vortex vortex patch. In particular the effects of nonlinear suppression of the parametric instability growth on a couple of examples. I believe the paper is interesting and overall well-written. I believe also that most of the results are original and can be accepted for publication with minor corrections (mostly typographical errors).*
*Minor points:*
*1) P5l l.10. 'omitting the fast-oscillating term ...' Why can the author do this? Does this term average to 0? Are the authors making a fast-time/slow-time separation?*
**Author's Response: Yes. The terms average to zero. The corresponding clarification has been added to the text.**

*2) Fig 3 and 4 should be more explained in the text, and caption should provide more information:*
*Questions which come to mind immediately:*
*i) In fig 3 and 4: are e=0.15 and \gamma = 0.02 from fig 1 still used? The same question goes for fig 2 in fact.*
*ii) Fig 3,4 a) Can the authors add a short sentence provides the details on how are in practice they obtained their Poincaré sections?*
*iii) Fig 3,4 b) What is the exact starting point of the trajectories used to illustrate the generic behaviour?*
*iv) It is unclear visually whether the trajectory in Fig 4b keeps spiraling outward for long times.*
**Yes. All the figures are for the same parameters e0=0.15, gamma0=0.02. More explanations have been added as follows: "To corroborate this effect, a Poincare section shown in fig. 3a is presented. To construct this one and all the following Poincare sections, we plot the position of a phase trajectory exactly in a perturbation period $2\pi/\nu$. Thus, a chaotic trajectory appears as a set of disorder points and a regular trajectories appears as a closed linked smooth orbit in the sections."**
**The trajectories start at the steady-state elliptic point $\varphi_0 = \pi/4, \varepsilon \approx 2.09244$ in all the figures. The corresponding clarification has been added to the text.**
**No, the trajectory spirals only until it reaches the region of high nonlinearity (chaotic region in the Poincare section in fig. 4a), then it spirals back. However, the parametric resonance results in significant change of the ellipse characteristics contrary to the case shown in fig. 3.**

*Minor points, typographical errors:*
*o) Abstract: add a full stop at the end of the first sentence.*
*a) p3, l1 "x-axis" -> "$x$-axis'*
**Corrected**
*b) p3, eqn (3) \Omega seems undefined in the present paper. The author should not expect the reader to read Bayly et al (1996) to understand symbols.*
**Should be $\gamma_0$ instead of $\Omega$**
*c) p5, l7 What is $\tau$. It is a rescaled time $t$ or just $t$?*
**Should be just $t$**

*d) p5, 16. The sentence unclear. Maybe rephrase as "...if the argument in the right- handed exponential function.." Then, on the next line, typo : "parametrci" -> "parametric"*

***Corrected***

*e) p6. l9 & l16. Be more specific when referring to "primary" and "secondary" zones. What is meant? I guess primary is the zone around \nu = 0.6 and the secondary the one around \nu = 0.3 but it is unclear.*

**The primary zone of the parametric instability is the widest zone located near $\nu=0.6$. The secondary zone is the one located near $\nu=0.3$.**

*f). p7 caption of figure 4: "The same as in fig. 3" (not 4).*

*g) p8. l10 typo " nonlnear" -> "nonlinear".*

**Thank you very much for pointing all these shortcomings.**

**We again thank the reviewer for the recommendations.**

Best wishes,
Eugene A. Ryzhov, Konstantin V. Koshel

---

## Author Comment (AC3) · 31 Oct 2016

Response to Reviewer 3's comments on the paper "Parametric resonance in the dynamics of an elliptic vortex in a periodically strained environment" by K.V. Koshel and E.A. Ryzhov.

We thank the reviewer for commenting on the manuscript. We have addressed the issues raised by the reviewer. The following is the point-by-point responses to the reviewer's list of questions:

*Reviewer's comments:*
*This work considers the stability for an elliptic vortex subjected to an external straining flow, examining different dynamic regimes depending on the external flow parameters. This is a novel work, very much of interest to and worthy of publication in this journal. I find no problem technically with the paper, only minor typos, as well as some sentences which should be changed to provide more clarity.*

*(1) page 1, Abstract, line 2: missing full stop after the word "regimes".*
*(2) page 1, line 15: Suggest changing wording to "Such models are usually highly nonlinear making it possible to gain insight into many phenomena that are difficult to predict within a geophysical setting"*
*(3) page 1, line 24: Suggest changing wording to "In the case of a stationary deformation, the elliptical vortex is able to perform..."*
*(4) page 2, line 3: Suggest to remove redundant phrase "not performing any motion", or else add comma , before the word not.*
*(5) page 3, line 9: Change to "these equations allow for parametric instability."*
*(6) page 3, line 11: Remove words "type of the"*
**Author's Response: Thank you very much for pointing these shortcomings. All has been corrected.**
*(7) page 3, line 14: Unclear what is the intended meaning of the line "provident time- dependent strain and rotation rates...", please clarify*
***Should be "given time-dependent ….."***
*(8) page 3, line 25: Typo, remove repeated line "The homoclinic separatrix...."*
*(9) page 4, line 13: change word "on" to "of"*
*(10) page 5, line 17: Typo, change "parametrci" to "parametric"*
*(11) page 6, line 6: add word "over" after prevail*
*(12) page 6, line 7: change "realizes" to "is realized"*
*(13) page 6, line 7: change "very" to "immediate"*
*(14) page 8, line 8: Typo, change "stead" to "steady"*

**All has been corrected.**

**We again thank the reviewer for the recommendations.**

Best wishes,
Eugene A. Ryzhov, Konstantin V. Koshel